# Distributed-Framework Basin Modeling System: II. Hydrologic Modeling System

Gang Chen [1,2], Wenjuan Hua [2], Xing Fang [3], Chuanhai Wang [1,2,*] and Xiaoning Li [2]

1 State Key Laboratory of Hydrology-Water Resources and Hydraulic Engineering, Hohai University, Nanjing 210098, China; gangchen@hhu.edu.cn
2 College of Hydrology and Water Resources, Hohai University, Nanjing 210098, China; huawenjuan0106@126.com (W.H.); xzl0938@hhu.edu.cn (X.L.)
3 Department of Civil Engineering, Auburn University, Auburn, AL 3684 9-5337, USA; xing.fang@auburn.edu
* Correspondence: chwang@hhu.edu.cn; Tel.: +86-136-0518-1550

**Abstract:** A distributed-framework hydrologic modeling system (DF-HMS) is a primary and significant component of a distributed-framework basin modeling system (DFBMS), which simulates the hydrological processes and responses after rainfall at the basin scale, especially for non-homogenous basins. The DFBMS consists of 11 hydrological feature units (HFUs) involving vertical and horizontal geographic areas in a basin. Appropriate hydrologic or hydraulic methods are adopted for different HFUs to simulate corresponding hydrological processes. The digital basin generation model is first developed to determine the essential information for hydrologic and hydraulic simulation. This paper mainly describes two significant HFUs contained in the DF-HMS for hydrologic modeling: Hilly sub-watershed and plain overland flow HFUs. A typical hilly area application case study in the Three Gorges area is introduced, which demonstrates DF-HMS's good performance in comparison with the observed streamflow at catchment outlets.

**Keywords:** hydrologic modeling; distributed-framework model; hydrological feature units; hilly area application

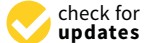



## 1. Introduction

Hydrological modeling is an important technique for flood forecasting and prevention, as well as water resource allocation. Over the past 50 years, various hydrological models have been promoted, tested, and applied. Hydrological modeling characterizes real hydrologic features and systems using small-scale physical models, mathematical analogues, and computer simulations for applications in flood prediction, hydrologic design, water resources planning and management, coupled systems modeling, etc. Hydrological models can be classified into four types: Metric models (essentially empirical and characterizing the system response from the available data based on observations), conceptual models (the model structure is specified prior to modeling and the model parameters may not have a direct physical interpretation), physics-based models (representing the component hydrological processes using the governing equations of motion based on continuum mechanics), and hybrid models (including elements of two or more above types) based on their model structure. Other types of classification are also commonly used, such as lumped models (treating the catchment as a single unit, with state variables that represent averages over the catchment area, expressed by differential or empirical algebraic equations) and distributed models (making predictions that are distributed in space, with state variables that represent local averages, by discretizing the catchment into a number of elements (or grid squares) and solving the governing equations for the state variables associated with every element) [1–4]. However, there is still a large demand for a model system to simulate all characteristic units (non-homogenous) of the hydrologic process at the basin scale.

Hydrological processes in areas with different conditions should be simulated with a set of proper models or computing methods, especially for multi-basin joint simulation in complicated structure areas [5]. Different from the hydrologic model system (HMS) (with four modules: Soil hydrologic model (SHM), terrestrial hydrologic model (THM), groundwater hydrologic model (GHM), and channel groundwater interaction (CGI)) developed by Yu [6], the distributed-framework basin modeling system (DFBMS) introduced in the first series paper [7] (Distributed-Framework Basin Modeling System: I. Overview and Model Coupling) consists of different hydrological feature units (HFUs) based on the runoff generation/concentration/routing mechanisms. The runoff generation is hydrological processes to remove rainfall losses from the gross rainfall to produce the rainfall excess. The runoff confluence is for the runoff from overland surfaces to gather/concentrate/converge to small streams or drainage channels. The runoff routing is for the runoff to travel along a river from upstream to downstream locations. Hydrologic and hydraulic modeling are two main aspects of a basin-scale hydrological modeling system, where the former usually refers to hydrological processes. The traditional hydrologic runoff concentration calculation method might have poorer performance than the hydraulic method in a distributed hydrological model, while recent hydrological models such as HEC-HMS (Hydrologic Engineering Center's Hydrologic Modeling System) and the Xinanjiang model [8] can provide good simulation results. Ahirwar et al. [9], for example, evaluated the performance of the Xinanjiang rainfall-runoff model for six watersheds. Moreover, in comparison to distributed models from the participants such as ARS (Agricultural Research Service), ARZ (University of Arizona), OHD (Office of Hydrologic Development), and HRC (Hydrologic Research Center), some simulations of the lumped model such as the Sacramento Soil Moisture Accounting (SAC-SMA) model can have better performance [10,11]. Therefore, various hydrologic calculation methods need to be efficiently applied in different modeling conditions. Several matured hydrologic models have been developed, such as TOPMODEL [12], as well as the HBV (Hydrologiska Byrans Vattenavdelning), SWAT (Soil and Water Assessment Tool), MIKE SHE (Systeme Hydrologique European), Xinanjiang, and VIC (Variable Infiltration Capacity) models, among others [13]. For practical application, the deterministic model is usually preferred, since it can give the sole output for one set of input values to support the hydrological forecast. TOPMODEL is a semi-distributed physical-based model [12] that is usually used in catchments with shallow soil and moderate topography [13]. HBV is a semi-distributed conceptual model [14] with different versions used in different regions and climatic conditions. MIKE SHE is a physical-based model that can be applied to a wide range of situations using parameters with physical interpretation and provides a large amount of simulation results [15]. SWAT is a complex physical-based model that is efficient and reliable in testing and forecasting water and sediment circulation and agriculture production [16]. However, adding complexity does not necessarily lead to improved hydrological model performance [17]. Xinanjiang is a conceptual model without complicated features such as soil moisture storage capacity [18], which is lumped or distributed in different versions. It is suitable for basins in humid and semi-humid areas and has been proven to have high efficiency [8,9,19,20].

Since hydrological simulation is complex and diverse, many hydrological models need to be built to satisfy different application situations, and the structure or division principle of the modeling system should first be considered and improved. In order to emphasize the hydrologic modeling system, this paper lists and briefly explains all the HFUs in the DFBMS, and then detailly explains those methods and models originally proposed or successfully implemented in the distributed-framework hydrologic modeling system (DF-HMS) so far. A proper simulating technology or method is adopted specially for each hydrological process in certain HFU. Hilly sub-watershed HFUs and plain overland flow HFUs are discussed in detail in this paper. Additionally, the digital basin generation model is first developed for DFBMS users to determine the essential information for hydrologic and hydraulic simulation. To present the practical application process and effectiveness of

this hydrologic modeling system, a typical hilly area case study in the Three Gorges Area is also given in this paper.

## 2. Materials and Methods

The hydrological feature units (HFUs) are the fundamental building blocks of the DFBMS. A HFU is an area/region in a basin/catchment that has the same mechanism of runoff generation and/or movement (including concentration and routing). According to different mechanisms of runoff generation and/or movement, the DFBMS includes 11 kinds of HFUs, which were also discussed in the first series paper [7]. Figure 1 presents the structure of the DFBMS, including snowfield, hilly sub-watershed, hilly river, plain overland flow, plain river, urban pipe network, lake and reservoir, hydraulic structure, karst region, unsaturated soil water, and saturated groundwater HFUs. The DFBMS includes 2 professional modeling systems: A distributed-framework hydrologic modeling system (DF-HMS) and a distributed-framework river modeling system (DF-RMS). The DF-HMS simulates various hydrological processes for each HFU with precipitation (rainfall or snowfall) as the model input. The DF-RMS is a professional modeling system for hydraulic modeling, which contains different HFUs to simulate the runoff movement through a system of rivers, river networks, storage units, lakes, and hydraulic structures [21]. For some large basins, both of these professional modeling systems may be applied coherently for basin hydrological simulations.

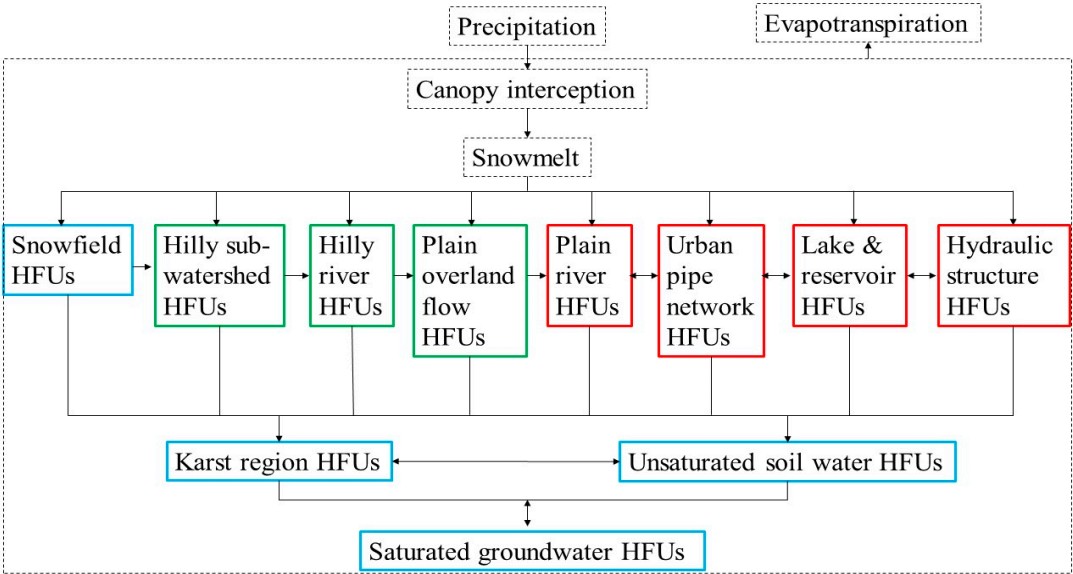

**Figure 1.** Structure of distributed-framework basin modeling system including different hydrological feature units or hydrological feature units (HFUs) and hydrologic processes.

Runoff generation was mostly simulated by hydrologic methods, while runoff movement, including confluence and routing, was simulated by both hydrologic and hydraulic modeling. Three HFUs (in green box in Figure 1) with hydrologic modeling are explored in detail in this paper. Four HFUs (in blue box in Figure 1) that were seldom used are described in less detail. Moreover, 4 HFUs (in red box in Figure 1) relating to hydraulic modeling, except for urban pipe network HFUs (which will be introduced in a future paper), were introduced in the third series paper [21], "Distributed-Framework Basin Modeling System: III. Hydraulic Modeling System," and are also given a brief explanation here. This paper is mainly focused on hilly sub-watershed and plain overland flow HFUs, which are currently the most widely used. For each HFU, the solving engine can be set for different simulation methods and module combinations, including partial distributed, lumped, conceptual, physical-based dynamic, and statistical models. That is, the appropriate model

can be applied according to different requirements. To consider spatial changes in the basin, HFUs can be further divided into hydrologic computing units (HCUs). A HCU is similar to the hydrological response unit, which is used as a discrete element for the simulation of a river basin. However, HCUs are not only employed for basin discretization in the DF-HMS. The main representations of HCUs include sub-watershed, sloped planes for overland flow, river cross-sections, and grids in DFBMS. For example, river cross-sections are HCUs for hilly river HFUs or plain river HFUs in 1-D simulations. Computational grids are HCUs for plain overland-flow plane HFUs or saturated groundwater zone HFUs in 2-D simulations. Moreover, each sub-watershed can be an HCU for the hilly watershed HFU. More details about HCU were explained in the first series paper [7].

In Figure 1, hydrologic processes are in dotted-line boxes and HFUs are in solid-line boxes. The HFUs are also contained in the large dotted-line box because other hydrologic processes can happen in a part of the HFUs depending on the circumstances, such as depression detention, infiltration, and runoff generation and movement. Some HFUs may change to other HFUs in special conditions such as extreme rainfall, flood events, seasonal variations, and managed diversion of a flood. For example, the flood districts are normally rain-fed low-lying lands that can be simulated as an unsaturated soil water HFU or saturated groundwater HFU. However, when the flood districts are used to store runoff overflowed from rivers and transfer flood discharge if necessary (during a flood event), then it should be simulated as lake and reservoir HFU. The double-sided arrows in Figure 1 are used between these HFUs. Precipitation was mainly obtained and processed based on observed and computed data from meteorological stations. Some calculation methods from other models, such as the SHE model, can be applied to hydrologic processes like evapotranspiration and canopy interception in these HFUs [15,22]. For example, constructing $CO_2$ cycle patterns [23] is one of the methods to build canopy interception and evapotranspiration models. In practice, it is usually processed by a simplified coefficient method (e.g., parameter K is the ratio of potential evapotranspiration to pan evaporation), especially in real-time flood forecasting. Snowmelt was considered in the snowfield HFUs. The related hydrologic method or model for snowmelt has not been developed in DFBMS, since all past applications of DFBMS have not needed to model the snowmelt process. However, the infrastructure for integrating snowmelt processes, such as the commonly used degree-day method [24], has been developed. For the lake and reservoir HFUs, the main input data are characteristics of the water body. For flood districts, paddy fields should be considered as the lake and reservoir HFUs when it is flooded. The zero-dimensional model and two-dimensional model [21] (described in the third paper) are available in DFBMS to make the necessary calculation. Table 1 lists the calculation methods and models integrated with the DFBMS so far. More details are given in relevant sections in this paper (originally proposed methods or used in the case study) or reference document (methods introduced in series papers or methods from other researchers). Although there are not abundant methods and models originally proposed or developed for all the HFUs (Figure 1), DFBMS also has preprepared structures for other models/programs/modules to integrate them through the dynamic library method. Therefore, some existing methods and models from other researchers were also recommended for different demands in this paper.

**Table 1.** Calculation methods and models integrated with the distributed-framework basin modeling system (DFBMS).

| Modeling Systems | Classification | HFUs | Methods or Models | Conditions | Inputs | Parameters |
|---|---|---|---|---|---|---|
| All | Geographic | All | Digital Basin Generation Model | With DEM data | DEM or GIS map | This paper Section 2.1 |
| DF-HMS | Hydrologic | Hilly sub-watershed | Xinanjiang model | Humid and semi-humid area | Measured precipitation, Pan evaporation | This paper Section 3.2 and [18] |
| DF-HMS | Hydrologic | Hilly sub-watershed | TOPMODEL | Based on digital Topography model | Topographic data, precipitation | [12] |
| DF-HMS | Hydrologic | Hilly river | Muskingum model | Confluence in river | Upstream inflow | This paper and [25] |
| DF-HMS | Hydrologic | Hilly river | Characteristic river length method | Hilly area | Inflow | [26] |
| DF-HMS | Hydrologic | Plain overland flow | Rainfall-runoff model in delta plain | Plain area | This paper Section 2.2.3 | This paper Section 2.2.3 |
| DF-RMS | Hydraulic | Lakes & reservoirs | Zero-dimensional model | 3rd paper | 3rd paper | 3rd paper |
| | | | Two-dimensional model | 3rd paper | 3rd paper | 3rd paper |
| DF-RMS | Hydraulic | Plain river | One dimensional hydrodynamic model | 3rd paper | 3rd paper | 3rd paper |
| | | | Two-dimensional hydrodynamic model | 3rd paper | 3rd paper | 3rd paper |
| DF-RMS | Hydraulic | Hydraulic structures | Gate and dam simulation | 3rd paper | 3rd paper | 3rd paper |
| Other | Hydraulic | Urban pipe network | Urban pipe network model | Further paper | Further paper | Further paper |

*2.1. Digital Basin Generation Model*

The digital basin generation model is the foundation of the DFBMS's geographic capabilities. It was developed based on the digital elevation model (DEM) as the input data source to determine the essential information for hydrologic and hydraulic simulation on the basin scale. Three DEM data structure types [27] are commonly used for hydrologic and hydraulic analysis: Raster DEM, triangulated irregular network (TIN), and elevation contours. Compared to TIN and elevation contours, the raster DEM data structure and the topological features are simple and can be quickly implemented by a numerical method with less computation cost. O'Callaghan and Mark [28] reported a technique for extracting major drainage path networks from raster DEM, and its performance appears to be consistent with the visual interpretation of drainage patterns from elevation contours. After that, more methods were promoted to develop a runoff concentration path model [29,30], catchment area [31], and drainage networks [32] using raster DEMs.

2.1.1. Runoff Concentration on Underlying Surface for Other Models

Single flow direction (SFD) algorithm and multiple flow direction (MFD) algorithm are 2 main types of routing algorithms. The SFD algorithm assumes that flow occurs only in the steepest downslope direction from any given point, while the MFD algorithm assumes that flow occurs in all downslope directions from any given point [33,34]. In this case, the MFD algorithm is not suitable for determining the dividing lines of watersheds and centerlines of rivers, which are essential data for building the DFBMS. Although the MFD algorithm seems to give superior results in the headwater region of a source channel, the SFD algorithm is superior in zones of convergent flow and along well-defined valleys [29,31]. Therefore, for all practical purposes, the SFD algorithm was adopted for the DFBMS described here. According to this concentration algorithm, all flow is directed into the neighboring cell corresponding to the highest gradient, and the flow from a single cell generally follows a unique path to the outlet [34].

The steps to develop the geographic and hydrologic parameters for the sub-watershed were based on the runoff concentration path across the landscape, as follows:

1. Input the raster DEM data for analysis.
2. Calculate the gradients between a cell and all lower neighboring cells to identify the steepest downslope for every cell and accumulate catchment area downslope along the runoff concentration paths connecting adjacent cells.
3. Determine the sub-watershed area and watershed dividing line based on the area threshold and runoff concentration path.
4. Generate the drainage, channel, and river network based on the runoff concentration path.
5. Calculate the geographic and hydrologic parameters for the sub-watershed, such as slope, slope orientation, and area.

One difficulty in generating the runoff concentration path is the measure dealing with the flow direction for the lowest cell (depressed pit), which exists in real watershed geometry or is caused by the DEM data. Different measures have been promoted in previous studies [28,35,36] to determine the flow direction for the lowest cell. O'Callaghan and Mark [28] presented a method that handles depressed pits introduced by DEM data smoothing to reduce the number of depressed pits generated by the data collection system. However, that method can only deal with small and shallow pits and considers pits on the land as errors in the elevation dataset and attempts to remove them. Band [35] used a method that increases the elevation of depressed pits until a downslope runoff concentration path to an adjacent cell becomes available, under the constraint that flow may not return to the depressed pits. Jenson and Domingue [37] used a method that fills each depressed pit in the DEM to the elevation of the lowest overflow point out of the sink, which subsequently modifies the flow direction in all flat areas to direct flow from each inflow cell on the perimeter of the depressed pit to the nearest outflow cell on the perimeter.

Martz and Jong [36] developed a modified elevation matrix to illustrate the topographic change caused by filling all depressed pits to the elevation of their overflow outlets. Subtracting the original from the modified elevation matrix gives the maximum depth of water ponding on the land surface. The local catchment area in that study was determined by stopping the flowline advance at depressed pits (those with no adjacent points at lower elevation), whereas global catchment was determined by allowing the flowline to advance through depressed pits via the lowest available outlet, and only stopping the advance when the edge of the elevation model was reached. As a byproduct of the analysis, a modified elevation matrix was produced, from which another relative position variable, the maximum depth of water ponding, could be determined easily for each point represented in the matrix.

Martz and Garbrecht [38] compared 10 algorithms to automate the identification of depressed pits and treatment from the DEM. In that study, they summarized that the basic difference between Jenson and Domingue's approach, and their approach lies in their implicit assumptions about how the land surface is modeled by the DEM [31]. Jenson and

Domingue [37] modified runoff concentration directions to impose a flow path on each depressed cell, implying that depressed pits are primarily data errors or artifacts. Martz and Jong changed the catchment area to simulate depressed cell overflow, implying that the topography is represented accurately and that any depressed cells are real topographic features that should be treated as small ponds or reservoirs. It follows that the selection of the most appropriate method for treating depressed pits should consider the accuracy and precision of the DEM to be analyzed.

Wang and Liu [39] presented an efficient method for identifying and filling depressed pits in a high-resolution DEM produced by airborne light detection and ranging (LiDAR). The technique can simultaneously determine runoff concentration paths and spatial partitioning of the watershed with 1 pass of processing. The method represents a novel concept of overflow elevation and the lowest-cost search for an optimal runoff concentration path.

### 2.1.2. Runoff Concentration in DFBMS

In the DFBMS, the fastest runoff concentration path (FRCP) with the D8 procedure was adopted to deal with depressed pits on surface DEM and runoff path generation. For the D8 method, the slope of a cell $K$ and 8 neighboring cells was calculated using Equation (1), and the runoff can flow along all positive slopes between cell $K$ and the 8 neighboring cells. Eight direction codes were used to indicate the runoff directions: 1, east; 2, southeast; 4, south; 8, southwest; 16, west; 32, northwest; 64, north; and 128, northeast (Figure 2A).

$$S = \frac{\Delta Z}{D} \tag{1}$$

Here, $S$ is the slope of cell $K$ and neighboring cells, $\Delta Z$ is the elevation difference of cell $K$ and neighboring cells ($Z_K - Z_{neighbor}$), and $D$ is the distance of cell $K$ and neighboring cells. The slope S is greater than 0 when elevations of neighboring cells are lower than elevation of cell K. Figure 2B shows an example of 4 cells (0–3), with 5 positive slopes (S01, S02, S03, S12, and S32) for us to discuss with respect to the FRCP method.

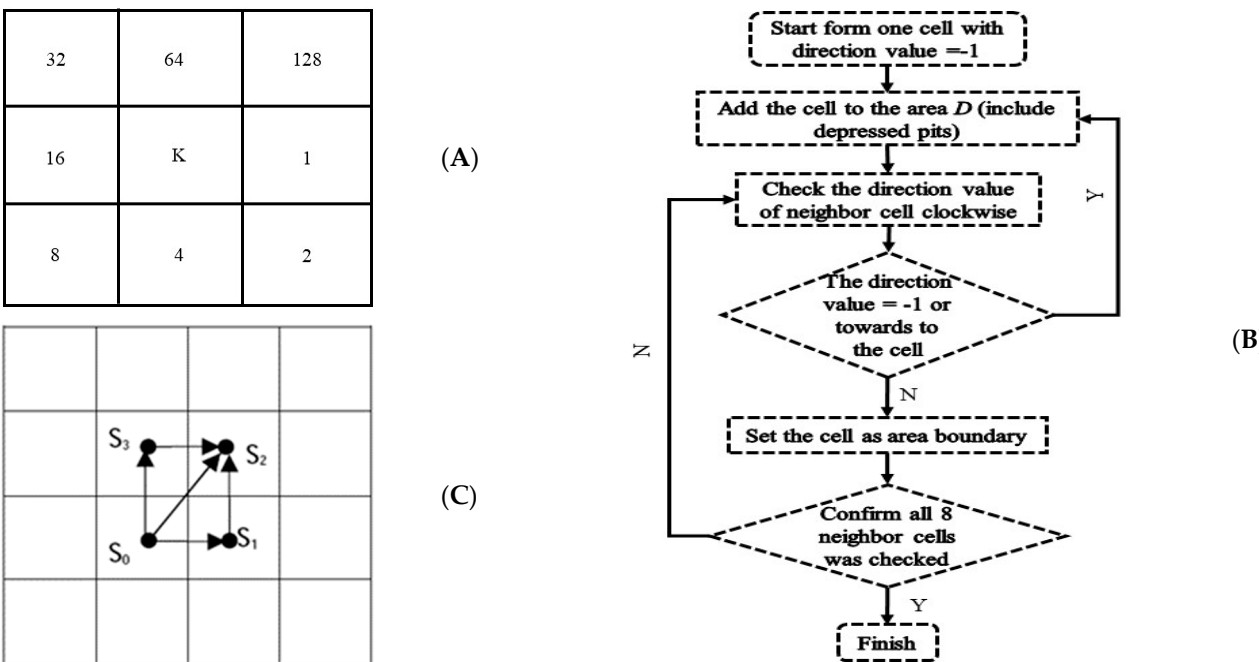

**Figure 2.** (**A**) Codes for runoff concentration directions. (**B**) Flowchart of area accumulation method for runoff concentration path generation. (**C**) Scheme of fastest runoff concentration path method involving with four cells (0–3) having 5 positive slopes ($S_{01}$, $S_{02}$, $S_{03}$, $S_{12}$, and $S_{32}$).

The following procedure was used to determine the runoff concentration path:

1.　We assigned 0 to the runoff concentration direction of invalid cells with DEM value equal to –9999.
2.　For cells at the boundary, 3 treatment methods were adopted for different situations: (a) The runoff concentration direction was set to be out of the domain when the maximum slope of the cell was smaller or equal to 0, (b) the runoff concentration direction was set to the maximum slope direction when there was only one maximum slope larger than zero, and (c) the runoff concentration direction was established based on Figure 2B when there was more than 1 slope larger than 0.
3.　For other cells, the slopes of cell $K$ and 8 neighboring cells were calculated first, then 4 treatment methods were adopted for different situations: (a) $-1$ was assigned to the cell to indicate the runoff concentration direction was undetermined when the maximum slope of the cell was smaller than zero (depressed pit cell), (b) the runoff concentration direction was set to the maximum slope direction when there was only 1 slope larger than 0; (c) the runoff concentration direction was set based on Figure 2B when there was more than one positive slope larger than zero, and (d) $-1$ was assigned to the cell to indicate the runoff concentration direction was undetermined when there was more than 1 slope equal to 0 ($Z_K = Z_{neighbor}$ for some neighboring cells).
4.　We started from the cell with runoff concentration direction value equal to $-1$ and checked whether the values of the 8 neighboring cells equaled $-1$ or not. After that, we generated the runoff concentration path with the area accumulation method, which is shown in Figure 2B.

Figure 2B shows the procedure of the area accumulation method, which include 4 steps: (1) Find the lowest and highest cells at the boundary of the depressed pit and flat areas $D$; (2) add a neighboring cell when the lowest cell elevation is the same as the highest cell elevation and repeat step 1; (3) assume the cell in the depressed pits and flat areas $D$, which is also the closest cell to the lowest cell at the domain boundary, is the outlet of the depressed pits and flat areas based on the FRCP method; and (4) modify and update the runoff concentration path direction for all cells lower than the outlet of the depressed pits and flat areas starting from the outlet cell. The runoff concentration velocity was estimated based on the Chezy equation (Equation (2)):

$$v = C\sqrt{RJ} \tag{2}$$

where $v$ is the runoff concentration velocity; $C$ is the Chezy coefficient; $R$ is the hydraulic radius, which is equal to the water depth ($H$) here; and $J$ is the water surface elevation slope.

In the fastest runoff concentration path method, it is assumed that the runoff will be concentrated along the path with the shortest travel time between the outlet and upstream cells in the depressed pit area.

Figure 2C shows an example of the FRCP method. The path from cell 0 to cell 2 was determined using the travel time calculated with Equation (3). Equation (3) was simplified with the assumption that the water surface elevation slope is the same for one depressed pit and flat area. Simplified Equation (4) was used in the code to find the fastest runoff concentration path with less computation effort. For the case shown in Figure 2C, the runoff will travel along the route with $t_{min} = \min(t_{S02}, t_{S01} + t_{S12}, t_{S03} + t_{S32})$.

$$t = \frac{D}{C\sqrt{\Delta HJ}} = K\frac{D}{\sqrt{\Delta H}} \tag{3}$$

$$t = \frac{D}{\sqrt{\Delta H}} \tag{4}$$

Here, $t$ is the time for the runoff to travel between 2 cells, $D$ is the distance between the centers of 2 cells, $\Delta H$ is the water depth difference of 2 cells, and $K = \frac{1}{C\sqrt{J}}$.

### 2.2. Primary HFUs of DF-HMS

In the DF-HMS, 2 typical HFUs were described in detail: Hilly sub-watershed and plain overland flow HFUs in non-homogeneous basins/catchments. In-depth explorations were done in the distributed-framework basin modeling system (DFBMS) for these HFUs, since they are the most common and frequently used in most regions. In most of study areas, hilly sub-watershed HFUs exist with hilly river HFUs, which will also be introduced in this section.

#### 2.2.1. Hilly Sub-Watershed HFUs

This kind of HFU is the most widely and deeply studied, and can be applied with many types of models, such as conceptual models (the Xinanjiang model), black-box models, TOPMODEL, and other distributed hydrological models [8,13,40]. Among those for hilly sub-watershed HFUs, conceptual models are the ones mainly adopted in practice because of their early development and wide application. Distributed models for this kind of HFU have been developed in recent years [11,13], highlighted by the application of 3S technology (remote sensing (RS), geography information systems (GIS), and global positioning systems (GPS)), especially DEM, which enables hydrological models to simulate the spatial distribution of the hydrologic process. However, many distributed models require significant input data and parameters, which causes a heavy workload and makes it hard to work with them in practice, especially for real-time prediction/forecasting problems.

As an example, the hilly sub-watershed HFUs were described using digital basin generation and a distributed Xinanjiang model in this study. In the first step, sub-watersheds were generated by the digital basin generation model, and their complete topological structure was stored to constitute the unit coupling topological structure model for the entire basin, such as the Three Gorges area (Figure 3). Then, the sub-watershed was further divided according to the temporal scale of discretization, where the temporal scale of discretization was the timestep to determine the length of the subchannel (discrete element of channel). Concretely, the channel in the sub-watershed was segmented into the subchannel by the flow path length, where the flow path length was the distance that water flows in river within the flow concentration time of 1 h. In most cases of implementing watershed hydrological simulation, the timestep of 1 h is used in hourly models for flood simulation and the timestep of 1 day is used in daily models for water balance studies. The timestep of most available measured data, such as rainfall, evaporation and discharge, was 1 h. Moreover, the timestep of input data and model computing of 1 h can both effectively reflect the hydrological process of the river basin and improve computational efficiency.

The catchment area of each subchannel is defined as the hydrologic computing unit of this kind of HFU, named the smallest runoff unit [40]. The entire HFU is divided by the smallest runoff unit, which is considered to be the best spatial discrete scale to match the temporal discrete scale, since it is the best condition for the Muskingum method when the time step is nearly equal to $K$ [25]. In the case in Figure 3, 79 sub-watersheds and 45 river segments were generated in the Three Gorge Reservoir area, which includes 79 hilly sub-watershed HFUs and 1 hilly river HFU (the reservoir).

Based on the partitioning of HFUs, we can compute the runoff generation and confluence processes for each smallest runoff unit, which are hydrologic computing units (HCUs) in this study. For the runoff generation process, the rainfall was calculated according to the input of measured data from precipitation station. For nonuniform rainfall, the basin can be divided into several polygonal areas by Thiessen polygon method [41] and the rainfall at the station stands for the average precipitation in each polygonal area. The runoff generation process simulation depends on the model selected in system for certain case study, which is not fixed. The Xinanjiang rainfall-runoff model [42] could be adopted as an example. The three-layer evapotranspiration pattern, the saturation excess runoff generation mechanism, and the three-component runoff division scheme are 3 main parts in Xinanjiang model for runoff generation. For the confluence process, the overland flow inside HCUs can be calculated through several methods, such as the linear reservoir or

geomorphic unit hydrograph method [43] of the Xinanjiang model. The drainage process of HCUs can be acquired, which then flows into the corresponding channel as the input of the hilly river HFUs.

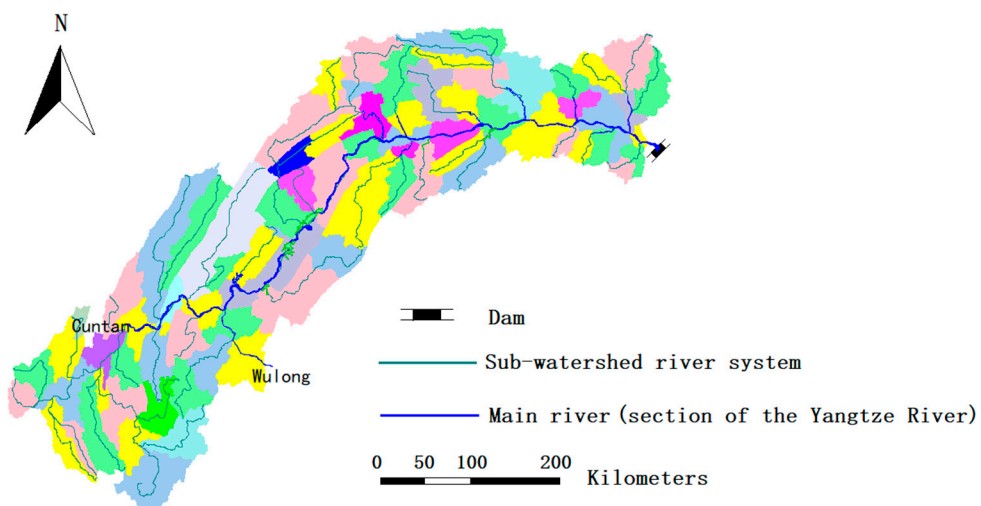

**Figure 3.** Sub-watersheds generated by the digital basin generation model in the Three Gorges area.

### 2.2.2. Hilly River HFUs

Hydrograph routing in the hilly river HFUs focuses on the drainage process of river cross-section. Since the river in the hilly area usually has a steep gradient and fast flow velocity affected mainly by gravity and frictional resistance, we can apply simplified dynamics methods to simulate the confluence process, such as the Muskingum or characteristic river length method [25,26].

The routing computation is from upstream to downstream step by step as the runoff on the slope converges into the corresponding tributary and then into the main river. Some model parameters related to geographic elements can be acquired from a digital basin generation model, such as the geomorphological unit hydrograph model [44].

### 2.2.3. Plain Overland Runoff Flow HFUs

The runoff generation/concentration mechanisms in the plain overland flow HFUs are quite different from them in the hilly area because of topographic characteristics, interaction with groundwater, and human activities. The Taihu basin (Figure 4), located in the delta plain in Jiangsu province in China, is an example case for the following discussion, which includes plain overland runoff generation HFUs and plain river HFUs. The overland flow in the delta plain usually has five main characteristics:

(a)  The delta plain is characterized by low elevation and flat topography, which leads to a slow confluence process and jacking influence between river flow and overland flow.

(b)  The delta plain maintains a shallow groundwater table, which causes more significant transformation between atmosphere water, surface water, soil water, and groundwater.

(c)  The underlying surface in the delta plain is more variable than in the hilly area, especially for specific land-use types. For example, paddy fields present different kinds of underlying surfaces in the growing and dormant periods, which means different rainfall-runoff mechanisms. Figure 4a shows an example of the land cover in the Taihu basin.

(d)  Extensive river networks are characterized by cyclic structure and uncertain catchment boundaries in the delta plain, which is far different from process of the dendritic river systems in the hilly area. The flow of the river here does not always converge

from the upstream to the downstream section, but rather moves in various directions. This open system with several outlets is hard to study by limited technology and conditions because it is difficult to measure flow and direction data of enough river control sections for model validation. Figure 4b shows an example of the plain river network in the Taihu basin.

(e)  Human activities in small hydraulic engineering projects such as polder embankments and water gates increase the complexity of the confluence process.

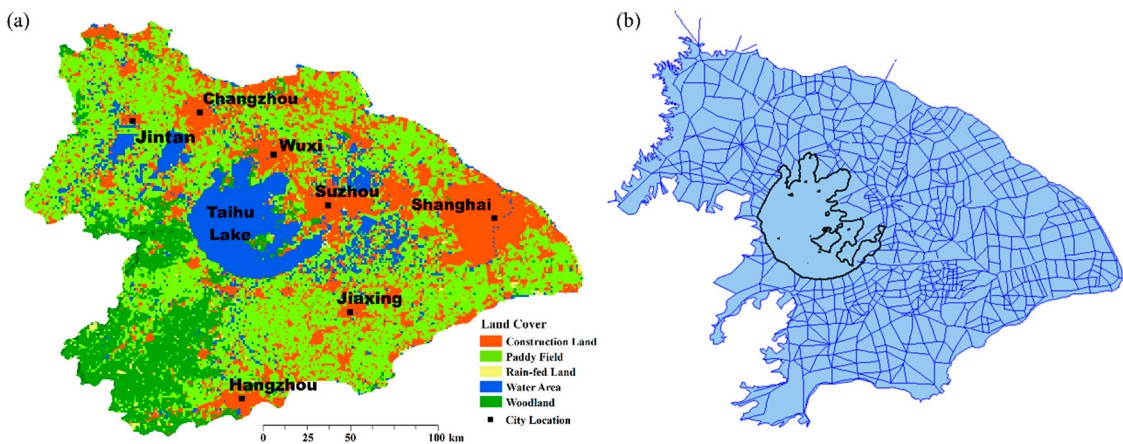

**Figure 4.** (**a**) Land cover and (**b**) river network in the Taihu basin.

Therefore, rainfall-runoff models for overland flow in the delta plain are generally required to solve problems in three main aspects: (1) Runoff generation on different underlying surfaces, (2) overland flow affected by human activities, and (3) distribution of outlets from overland flow to river network (distributed overland flow). Thus, related models are proposed in this paper to simulate runoff in the delta plain more reasonably.

Although the land use type is the most significant factor of the underlying surface, it is unnecessary to classify the land according to the classification system of national land use status, because it is too detailed to acquire related measured data to calibrate the model. Considering both availability and effectiveness, we divided the underlying surface into four categories: Water area, paddy field, rain-fed land, and construction land, which can be improved in a further study. A separate runoff generation model was built for each land category.

i.  Modeling in the water area

The water yield per unit area is almost equal to net rainfall in the water area, which can be calculated by the difference between rainfall and evaporation in each period as follows:

$$R_W = P - E \tag{5}$$

where $P$ (mm), $R_W$ (mm), and $E$ (mm) are rainfall, runoff depth, and evapotranspiration, respectively, within a period. The evaporator method is used to estimate daily evapotranspiration:

$$E = \beta \cdot E_C \tag{6}$$

where $\beta$ is the converting coefficient of the evaporator (e.g., evaporation pan), which can be checked from the related chart, and $E_C$ (mm) is the evaporation measured by the evaporator.

ii.  Modeling in rain-fed land

The Xinanjiang model, with a specific parameter range [45,46] for the delta plain, was used to simulate rainfall-runoff in rain-fed land. Since the climate in the delta plain is characterized by humid and subhumid areas, the Xinanjiang model [18] is an applicable hydrologic model. The range of parameters in the Xinanjiang model are listed in Table 2.

**Table 2.** Parameters in the Xinanjiang model.

| Number | Parameter | Definition | Range |
|:---:|:---:|:---:|:---:|
| 1 | K | Ratio of potential evapotranspiration to pan evaporation | 0.1–1.5 |
| 2 | UM | Upper layer soil water storage capacity | 5–20 mm |
| 3 | LM | Lower layer soil water storage capacity | 60–90 mm |
| 4 | C | Deep evaporation coefficient | 0.09–0.3 |
| | $W_{mm}$ | Maximum watershed soil water storage capacity (mm) | 70–210 |
| | B | the exponent of soil water storage capacity curve | 0.05–0.4 |
| 5 | SM | Free water storage capacity (mm) | 1–50 |
| 6 | EX | the exponent of soil water storage capacity curve | 1.0–1.5 |
| 7 | KI | Outflow coefficient of free water storage to interflow | 0.2–0.6 |
| 8 | KG | Outflow coefficient of free water storage to Groundwater | 0.2–0.6 |
| 9 | CS | Recession constant of surface runoff | 0.01–0.4 |
| 10 | CI | Recession constant of interflow | 0.1–0.99 |
| 11 | CG | Recession constant of groundwater runoff | 0.7–0.99 |
| 12 | KE | Routing time in channel unit (h) | 0–1 |
| 13 | XE | A weight factor of Muskingum method | 0–0.5 |

iii.    Modeling in the paddy field

The modeling of runoff generation in the paddy fields in China can be divided into six periods. The first one is the period before the growing period, when the mechanism of runoff generation is the same as on rain-fed land. Thus, the runoff yield model for rain-fed land introduced above was applied. The second period is the rice seedling bed period, which can be further divided into soaking and seedling periods. For the former, irrigation water and precipitation volume include both the part used to saturate the soil, recharge the groundwater, and produce regression flow, and another part to create certain water depth for seeding the field. For the latter, irrigation water and precipitation are mainly used to maintain the water depth in the paddy field, and the part of regression flow is counted as the runoff yield of the field into the river. The part of the paddy field not used for seeding in this period is still processed as rain-fed land. The third period is the primary growing period, which contains soaking and various growing periods (Table 3). When switching to this period, both the water deficit to saturation and soaking water volume should be considered in the water balance. The fourth period is the field drying period, when both the water storage on the field surface and the water volume between soil saturation moisture content and soil moisture content under saturated soil storage will be discharged by drainage. During this period, since no water is stored on the field surface, the initial soil moisture content is the soil moisture content under saturated soil storage. The water requirement for paddy growing is from soil storage, and the lower limit of soil moisture content for the paddy is assumed to avoid an unreasonable water loss calculation. The fifth period is the mature later period, which is processed as the field drying period. The last one is the period after the paddy is grown, when the soil moisture content depends on what was calculated in the mature later period. After that, the model for rain-fed land will be applied again.

**Table 3.** Irrigation program of paddy fields (example in the Taihu basin).

| Growing Period | Duration | Resistance to Submergence Depth (Hp) (mm) | Upper (Hu) (Lower (Hd)) Limit (mm) | Water Requirement Coefficient | Daily Leakage (mm) | Period of Paddy Field |
|---|---|---|---|---|---|---|
| Soaking | 16–25 May | 40 | 10 (5) | 1.00 | 2 | Rice seedling bed |
| Seedling | 26 May–13 June | 30 | 20 (10) | 1.00 | 2 | |
| Soaking | 14–23 June | 40 | 10 (5) | 1.00 | 2 | Primary growing |
| Resume growth | 24–30 June | 50 | 30 (20) | 1.35 | 2 | |
| Tiller earlier | 1–4 July | 50 | 30 (20) | 1.30 | 2 | |
| | 5–9 July | 0 | 0 (0) | 1.30 | 0 | Field drying |
| | 10–19 July | 50 | 30 (20) | 1.30 | 2 | Primary growing |
| Tiller later | 20–23 July | 50 | 30 (20) | 1.30 | 2 | |
| | 24 July–4 August | 10 | 0 (0) | 1.30 | 0 | Field drying |
| Booting | 5–18 August | 50 | 40 (30) | 1.40 | 2 | Primary growing |
| | 19–23 August | 0 | 0 (0) | 1.30 | 0 | Field drying |
| | 24 August–3 September | 50 | 40 (30) | 1.30 | 2 | Primary growing |
| Heading | 4–16 September | 50 | 30 (20) | 1.30 | 2 | |
| Mature | 17 September–15 October | 20 | 10 (0) | 1.30 | 0.7 | |
| | 16–20 October | 0 | 0 (0) | 1.05 | 0 | Mature later |

The irrigation program for the growing period of paddy fields in the Taihu basin is shown in Table 3. The growing period was divided into nine specific subperiods according to the vegetation process and requirements of rice. Different periods of modeling runoff generation in the paddy field can be repeated several times depending on the vegetation process, but field drying always follows primary growing in each subperiod. The related calculation method of runoff yield in the paddy growing period is described in detail.

The water depth of the paddy field $H$ can be calculated according to the water balance during each period:

$$H = H_0 + P - \alpha \cdot E - f \tag{7}$$

where $H_0$ (mm) and $H$ (mm) are the initial and ending water depth of the paddy field, respectively; $\alpha$ is the water requirement coefficient, which is defined by the ratio of water requirement to pan evaporation during the same period; and $f$ (mm) is the daily leakage of the paddy field.

According to multiple factors, such as the water requirement, suitable upper and lower limits of water depth, and resistance to submergence depth, for one block of paddy field with the same water depth, the runoff generation depth $R_r$ was calculated in each period:

$$R_r = \begin{cases} H - H_p & H_p < H \\ H - H_u & H_u < H < H_p \\ H - H_d & H_d < H < H_u \\ 0 & H < H_d \end{cases} \tag{8}$$

where $H_p$ (mm), $H_u$ (mm), and $H_d$ (mm) are the resistance to submergence depth, suitable upper limit, and lower limit, respectively, in different periods.

Overall, the calculation of runoff yield in a period depends on the principle of irrigation and drainage. For paddy fields in basins or subareas, the beginning and end date of

each growing period is different, and the water depth in a subarea is uneven. Therefore, $R_r$ in different blocks of paddy field for a subarea was classified as $R_1$, $R_2$, and $R_3$, as follows:

$$\begin{cases} If\ H = H_{min}\ and\ H < H_d\ , & then\ R_1 = \frac{H-H_u}{3} & until\ H_{min} = H_u\ ; \\ If\ H = H_{max}\ and\ H > H_d\ , & then\ R_3 = \frac{H-H_p}{3} & until\ H_{max} = H_u\ ; \\ \quad If\ \ H_d < H < H_p\ , & then\ R_2 = 0 \end{cases} \tag{9}$$

$$R = R_1 + R_2 + R_3 \tag{10}$$

where $R$ is the total water yield depth in a subarea during each period. Runoff is produced when $R$ is positive, and irrigation water volume is calculated when $R$ is negative.

iv.     Modeling in construction land

The underlying surface of construction land (urban and suburban areas) can be clarified as three types according to runoff generation: Pervious surface, impervious surface with depression detention, and impervious surface without depression detention. Pervious surfaces mainly consist of green belts in cities, which take up the proportion of $A_1$ (Figure 5) in construction land areas. Impervious surfaces with depression detention include pits, waterways, and pipe networks such as roads, roofs, and so on, which occupy $A_2$. Impervious surfaces without depression detention account for $A_3$. The model for construction land is generalized as a diagram in Figure 5.

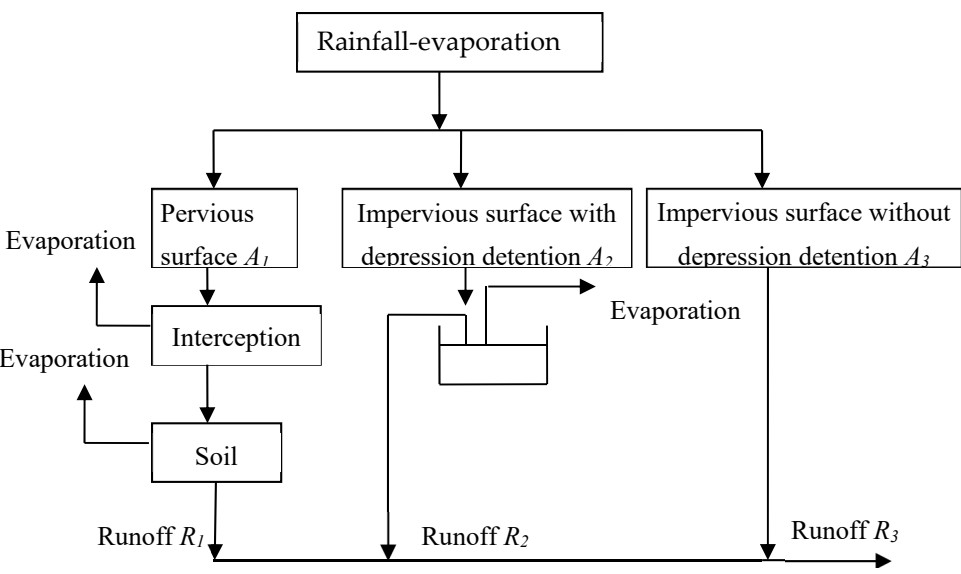

**Figure 5.** Logic structure diagram of construction land runoff generation model.

Runoff generation for pervious surfaces considering interception and soil evaporation was calculated using Equations (11) and (12):

$$F_r = \begin{cases} S_{e2} - S_{eL} & S_{e2} > S_{eL} \\ 0 & 0 < S_{e2} < S_{eL} \\ 0 & S_{e2} < 0 \end{cases} \tag{11}$$

$$S_{e2} = S_{e1} + P_E = \begin{cases} S_{eL} & S_{e2} > S_{eL} \\ 0 & S_{e2} < 0 \end{cases} \tag{12}$$

where $P_E$ (mm) is efficient rainfall considering potential evapotranspiration, $F_r$ (mm) is efficient rainfall on pervious surfaces after interception, $S_{eL}$ (mm) is the maximum

interception, and $S_{e1}$ (mm) and $S_{e2}$ (mm) are initial and ending interception in a period, respectively. Soil evaporation $E_s$ is subtracted from $F_r$ to calculate runoff $R_1$:

$$R_1 = F_r - E_s. \tag{13}$$

Runoff generation of impervious surfaces with depression detention was calculated based on water balance according to the initial and ending storage of depressions of each period step by step:

$$R_2 = \begin{cases} S_{t2} - S_{tL} & S_{t2} > S_{tL} \\ 0 & 0 < S_{t2} < S_{tL} \\ 0 & S_{t2} < 0 \end{cases} \tag{14}$$

$$S_{t2} = S_{t1} + P_E \tag{15}$$

$$S_{t2} = \begin{cases} S_{tL} & S_{t2} > S_{tL} \\ 0 & S_{t2} < 0 \end{cases} \tag{16}$$

where $R_2$ (mm) is the runoff depth on the impervious surface with depression detention, $S_{tL}$ (mm) is the maximum storage of the depression, and $S_{t1}$ (mm) and $S_{t2}$ (mm) are initial and ending storage of the depression during a period, respectively. The runoff generation model for impervious surfaces without depression detention is as follows:

$$R_3 = \begin{cases} P_E & P_E \geq 0 \\ 0 & P_E < 0 \end{cases} \tag{17}$$

where $R_3$ (mm) is the runoff depth on the impervious surface without depression detention. Thus, the total runoff depth $R$ (mm) on construction land can be obtained:

$$R = A_1 \cdot R_1 + A_2 \cdot R_2 + A_3 \cdot R_3. \tag{18}$$

2.2.4. Distributed Overland Flow Concentration Model in the Delta Plain

Two main aspects should be considered after overland runoff generation simulation: Spatial allocation of runoff from overland to the river network and the runoff confluence process with time. That is, these two issues of spatial and temporal distribution need to be solved.

Figure 6a is a sketch map of subareas separated by generalized river channels in the plain area. Those subareas are surrounded by the river reach, which is called the river-network polygon. For example, a closed polygon is formed by reaches 1, 2, 3, 4, and 5 (Figure 6b). The river-network polygon is divided into computational grids to calculate the runoff generation. The runoff in each grid flows to a certain river, which is the surrounding reach for this river-network polygon. A digital basin generation model was also applied here to build the catchment areas similar with hilly areas. For example, the river-network polygon generated five catchment areas according to five surrounding reaches (Figure 6b), where each colored catchment area corresponds to one river, indicating that the runoff of all grids in this catchment area contributes to that river. In addition, the proportion of each kind of underlying surface in each grid can be acquired according to the geographic information layers of four kinds of underlying surfaces: Water area, rainfed land, paddy field, and construction land. Then, the runoff process of each kind of underlying surface flowing into a certain river reach can be computed by space superposition operation of GIS. The linear reservoir method, the unit hydrograph method, and other methods can be applied to calculate the overland runoff confluence.

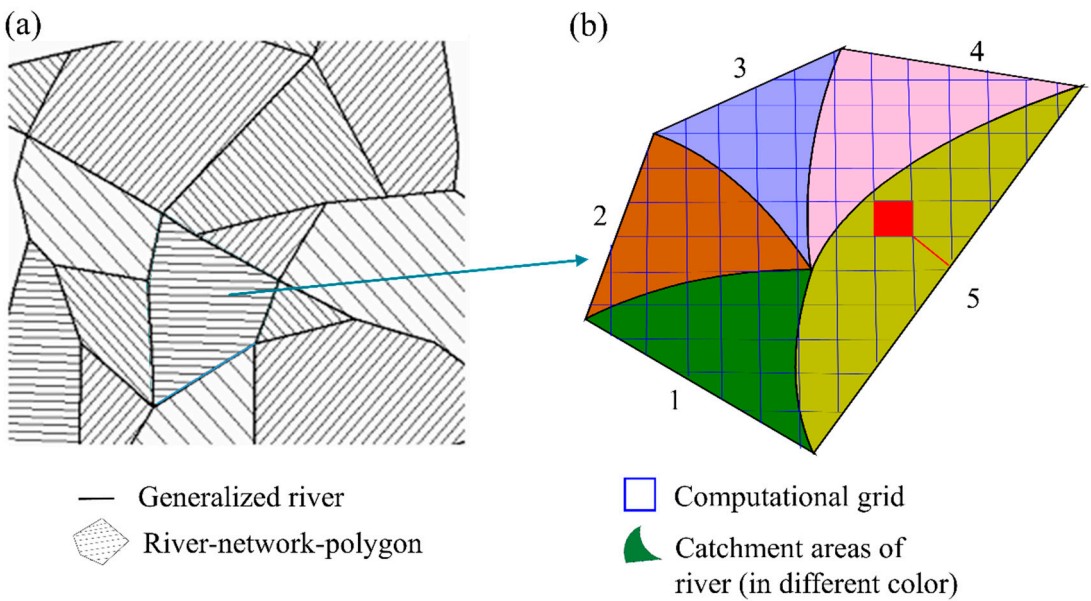

**Figure 6.** (**a**) Sketch map of division of overland area in plain area; (**b**) diagram of river-network polygon catchment areas of rivers.

### 2.3. Typical HFUs in Hydraulic Modeling System

In this section, a brief introduction is given as a guide to the third series paper "Distributed-Framework Basin Modeling System: III. Hydraulic Modeling System" [21]. In the distributed-framework river modeling system, three kinds of HFUs are mainly used in DF-RMS:

(1). The plain river HFU, which is a confluence/routing hydrological feature unit. It includes rivers characterized by low flow rate, jacking influence between main streams and tributaries, and even reciprocating flow existing in the littoral region. Both the mean water level and discharge process and the distribution of water level and flow velocity in the river are sometimes considered. Therefore, the dynamical method was applied to deal with these confluence processes. The river flow model includes one-dimensional and two-dimensional models to simulate for single channels and the river network as well as the coupling process between different models.

(2). The lake and reservoir HFU, which is also a confluence hydrological feature unit. Lakes and reservoirs usually maintain a certain level of water, and water moves inside flood districts and polders after a dam breaks or a gate opens. Once the water flow exists, it will follow the same rule of confluence that was classified as the same HFU in the DFBMS. According to the accuracy requirement, the confluence simulation for this kind of HFU can be divided into three types: Zero-dimensional, two-dimensional, and three-dimensional models. However, the three-dimensional model is not commonly used at the basin scale.

(3). The hydraulic structure HFU, which is part of the confluence hydrological feature unit. It includes hydraulic structures such as gates, dams, reservoirs, and flood area entrance gates. This kind of HFUs is commonly applied in flood control/forecasting and water resource scheduling. Models for these HFUs focus on simulating the flow control and scheduling process.

### 2.4. Uncommon HFUs in Distributed-Framework Professional Model System

Moreover, uncommon HFUs are also included in the distributed-framework professional model system. Some of these HFUs only appear in a specific climate or environment, like snowfield and karst region HFUs. Models for snowfield HFUs generally include the energy-balance method and the diurnal temperature-index method [47,48]. Karst region

HFUs belong to the confluence type, with two main characteristics. On the one hand, there are several underground rivers, constructing a three-dimensional serried network. On the other hand, water exchange is generated between underground rivers and surface runoff, interflow, and groundwater. Both non-pressure and pressure flow exist and alternate. As a manmade product, urban pipe network HFUs involve hydraulic methods under limited conditions, which will be introduced in a further study.

Some HFUs rarely exist independently, such as unsaturated soil water and saturated groundwater HFUs. As a vertical component, unsaturated soil water HFUs usually exist with other HFUs and play an important role in runoff generation. The water deficit of unsaturated soil water HFUs must be considered for regions controlled by runoff yield under saturated storage. Dynamical methods are often used in saturated groundwater units. These methods can be accomplished by solving partial differential equations of groundwater motion numerically, for which the technology is comparatively mature in related commercial software, such as FEFLOW [49].

### 3. Case Study

*3.1. Study Area and Corresponding Hydrological Characteristic Unit*

This case study was an application in a hilly area, focusing on hydrologic modeling. Another case for the runoff generation application in the plain area was presented in the fourth series paper [50], which was a comprehensive simulation for runoff generation and movement in the river network area.

The Three Gorges area includes a 658 km-long section of the Yangtze River flowing from Zhutuo to Yichang, shown in Figure 7a, and 56,126 km$^2$ of surrounding subbasins (Figure 3). The floods in the Three Gorges area consist of three parts: 81% from the Yangtze River upstream above Cuntan, 11% from the Wujiang River upstream above Wulong, and 8% from the Three Gorges area. A drainage map, available rainfall stations, and hydrological stations (measuring streamflows) in the subbasins are presented in Figure 7b. Four representative hydrological stations are in red in Figure 7b, which are used to present in flood hydrographs for discharge simulation results in 2005 and 2006.

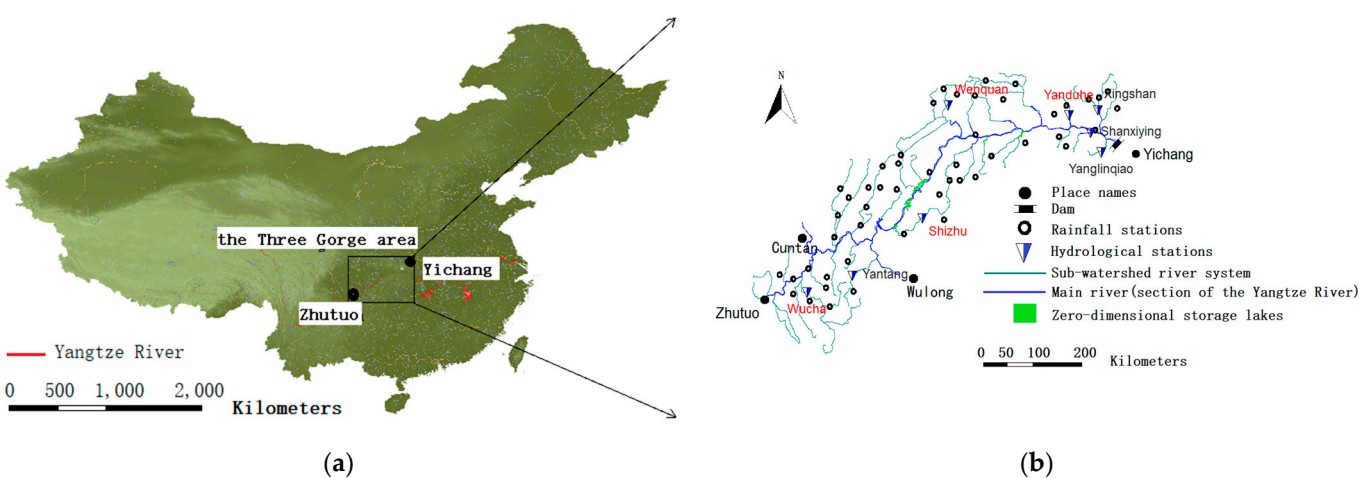

(**a**)　　　　　　　　　　　　　　　　　(**b**)

**Figure 7.** (**a**) Location of the study area and (**b**) drainage map of Three Gorges area.

According to the HFU concept in the DFBMS, the Three Gorges area can be generalized as two hydrological feature units: Hilly sub-watershed and plain river HFUs. The plain river HFUs exist not only in the plain area but also in the hilly area when the water flow characteristic accords with the corresponding HFU. Influenced by the large reservoir area, the hydraulic gradients along the reservoir were smaller, so the reservoir was recognized as one plain river HFU.

### 3.2. Methodology

For the case study, 79 sub-watersheds (Figure 3) (i.e., hilly sub-watershed HFUs) and 45 river segments (i.e., hydrologic computing units for the plain river HFU) were generated by the digital basin generation model for the Three Gorges area (Figure 8).

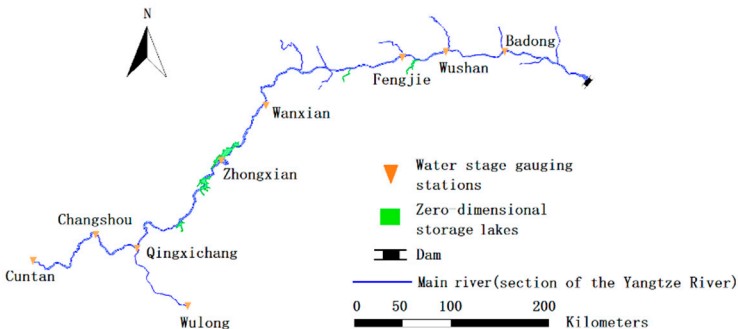

**Figure 8.** Generalized the main river network in the Three Gorges area.

For the hilly sub-watershed HFUs, the Xinanjiang model was used in each sub-watershed in this study, since a large number of studies have verified its applicability, especially in China [20,51,52]. The rainfall was calculated according to measured data from precipitation stations. Thiessen polygons [41] were generated to further divide the sub-watershed: The rainfall of each station was associated with a unique polygon. The Xinanjiang model [53] used in this case study contained a three-layer evapotranspiration pattern ($E_U$, $E_L$, $E_D$), saturation excess runoff generation mechanism, three-component runoff division scheme ($R_S$, $R_I$, $R_G$), and runoff routings in river channels (Muskingum method). More details about the Xinanjiang model are available in the literature [52]. In this case study, the outflow of each sub-watershed went directly into the connected main river. The input data were the measured rainfall $P$ and the pan evaporation $E_0$. Figure 9 shows the structure of the Xinanjiang model [53], and the parameters of the model are described in Table 2. In the daily model for hydrologic budget, the computational time is 1 year, and the computational timestep is 1 day. In the hourly model for flood forecast, the computational time depends on certain rainfall or flood event, and the computational timestep is 1 h.

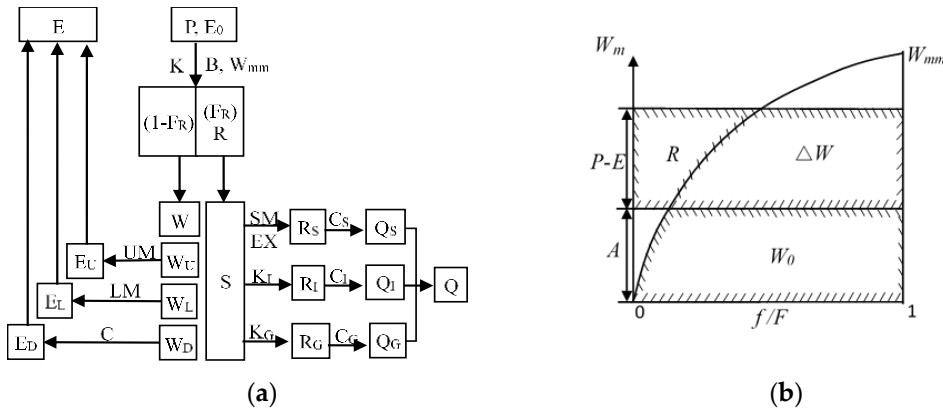

(**a**)  (**b**)

**Figure 9.** The structure of the Xinanjiang model without the impermeable area (**a**) and the diagrammatic sketch of the saturation excess runoff generation mechanism based on soil water storage capacity curve (**b**).

For the plain river HFU representing the Three Gorges reservoir, the simulated outflow from the reservoir was calculated using simulated inflows and measured real-time water levels just upstream of the dam (considering the reservoir bathymetry). These inflows are

equal to the simulated outflows from surrounding subbasins (Figure 3) using the hilly sub-watershed HFUs. The observed outflows from the reservoir were obtained from the hydrological station downstream of the dam and used to validate the simulation results. In addition, the reservoir regulation did not affect the simulation efficiency of this method, since its influence was taken into account by the measured water levels.

The generalized main river network, including the water stage gauging stations, is shown in Figure 8. The one-dimensional hydrodynamic model was used here for river flood routing. The upper boundary condition was the water level at Cuntan and Wulong, and the lower boundary condition was the measured water level at the dam. The main parameter was the roughness factor. Generally, the storage of the generalized river network was slightly smaller than the actual storage of the channel reservoir, so small branches and low-lying lands were generalized as zero-dimensional storage reservoirs to correct the error. Then, the relationship between the water level and the storage was obtained.

To achieve the aim of this paper and avoid redundant content in other series papers, the methods of relevant hydrodynamic model were described in the third series paper in detail.

### 3.3. Model Calibration and Validation

In this case study, the daily data from 2004 to 2008 were used for calibration of hydrologic budget. The available hourly data of flood events during the time period 2004–2005 and 2006–2008 were respectively used to calibrate and validate the flood forecasting model by the shuffled complex evolution (SCE-UA) optimization algorithm method [54]. The Xinanjiang model was used as the flood forecasting model. The study was focused on flood discharge, and the rainfall time step was 1 h. Flood event data from eight hydrological stations from the nonuniform year were adopted to do the prior calibration and validation of the reservoir inflow flood model, since rainfall and runoff were asynchronous in those basins in the Three Gorges area.

Relative error (*RE*, %), root-mean-square error (*RMSE*), and Nash–Sutcliffe efficiency (*NSE*) coefficient were used to assess the performance of the basin modeling system. The *RE* provides an evaluation of the flood peak discharge, the *RMSE* provides a general evaluation of the water balance, and the *NSE* is used to assess the efficient of fit between the simulated and the observed discharges. *RE*, *RMSE*, and *NSE* are calculated as follows:

$$RE = \frac{\sum_{i=1}^{n}(Q_{sim,i} - Q_{obs,i})}{\sum_{i=1}^{n} Q_{obs,i}} \times 100\% \tag{19}$$

$$RMSE = \sqrt{\frac{\sum_{i=1}^{n}(Q_{obs,i} - Q_{sim,i})^2}{n}} \tag{20}$$

$$NSE = 1 - \frac{\sum_{i=1}^{n}(Q_{obs,i} - Q_{sim,i})^2}{\sum_{i=1}^{n}(Q_{obs,i} - \overline{Q_{obs}})^2} \tag{21}$$

As presented in Tables 4–6, "Flood event" gives eight digits (e.g., 20070726 for Yantang station) for the year, month, and date when the flood happened. Rainfall is the average rainfall from representative precipitation stations according to the Thiessen polygon method. The comparison of observed and simulated flood peak discharge shown in Table 4 indicates that a relatively larger error of flood peak discharge happened in Wucha station. Three-fifths of simulation results of flood peak discharge in Wucha had errors over 10% during the study period, maybe due to the different rainfall characteristic and the different condition of soil water storage in the early stage, since the rainfall was calculated by the Thiessen polygon method but Wucha station is near the upper boundary of this large basin. Another station with a relatively large flood peak discharge error of −13.7% is Yanduhe, where there may be a measurement error because the observed data have a sudden and sharp increase in the peak.

**Table 4.** Rainfall, observed and simulated discharges, and simulation efficiency of 21 flood events at 8 hydrological stations.

| Hydrological Station | Flood Event | Rainfall (mm) | Observed (Simulated) Flood Peak Discharge (m³/s) | Flood Peak Discharge RE (%) | RMSE (m³/s) | NSE |
|---|---|---|---|---|---|---|
| Shanxiying | 20080625 | 254.4 | 166 (166.6) | 0.36 | 14.1 | 0.9 |
| Shizhu | 20050404 | 75.5 | 238 (243.6) | 2.4 | 16.3 | 0.8 |
| | 20060703 | 107.9 | 1091 (992.5) | −9.0 | 40.7 | 0.9 |
| Wenquan | 20040828 | 285.5 | 2033 (2077) | 2.2 | 95.9 | 0.9 |
| | 20050625 | 86.8 | 410 (377.9) | −7.8 | 28.7 | 0.9 |
| | 20050930 | 154.4 | 619 (580.1) | −6.2 | 45.6 | 0.9 |
| | 20060625 | 186.3 | 710 (719.4) | 1.3 | 48.3 | 0.8 |
| Wucha | 20040401 | 63.7 | 710 (725.6) | 2.2 | 71.4 | 0.7 |
| | 20040529 | 63.2 | 2290 (2524) | 10.2 | 215.2 | 0.9 |
| | 20050820 | 49.8 | 881.4 (786.9) | −10.7 | 73.6 | 0.9 |
| | 20060530 | 56.2 | 1085 (1232) | 13.5 | 112.4 | 0.7 |
| | 20081022 | 194.8 | 2360 (2553) | 8.2 | 191.4 | 0.8 |
| Xingshan | 20070616 | 215.6 | 624 (668.5) | 7.1 | 154.1 | 0.6 |
| | 20080415 | 88.7 | 345.1(326.3) | −5.4 | 26.0 | 0.9 |
| | 20080812 | 130.2 | 580.2 (559.4) | −3.6 | 56.9 | 0.8 |
| Yanduhe | 20050406 | 67.4 | 41 (40.9) | −0.2 | 3.6 | 0.9 |
| | 20060419 | 99.9 | 83 (71.6) | −13.7 | 6.6 | 0.9 |
| Yanglinqiao | 20060902 | 74.6 | 100 (96.3) | −3.7 | 7 | 0.9 |
| | 20080415 | 99 | 107.2 (110.4) | −2.9 | 9.1 | 0.8 |
| Yantang | 20070712 | 80.1 | 472 (462.6) | −2.0 | 36.8 | 0.9 |
| | 20070726 | 73.8 | 253.5 (256.4) | 1.1 | 11.1 | 0.9 |

Overall, the simulation efficiency evaluated using NSE and all observed discharges for each flood event were stable and reliable. The simulation results of discharge in eight subareas with hydrological stations generally show good performance. Only 3 of 21 flood events had an NSE of discharge simulation for flood events smaller than 0.8, and 13 events had an NSE larger than 0.9. The only flood with NSE smaller than 0.7 was flood 20070616 in Xingshan station, where the rainfall was large and the station was near the dam, so its flood receding process was slowed by the high water level in the river reservoir. However, the final result of discharge of outflow from the reservoir can be simulated better by coupling the result with the hydraulic method used in the river reservoir.

The results of discharge simulation in 2005 (Figure 10) and in 2006 (Figure 11) for four representative hydrological stations (in red in Figure 7) are presented in flood hydrographs. In most flood cases, the simulated flood peak discharges were lower than the observed ones, except for Wucha station. Generally, there was a large discharge, but relatively small rainfall measured at Wucha station, which suggests that other sources may recharge this watershed, such as the underground macropore flow.

The parameters of the hydraulic model were calibrated using the SCE-UA method. We used 22 flood events in 2007 and 28 flood events in 2008 as calibration and validation periods, respectively. The flood peak level and NSE were calculated, and the results of the hydraulic model calibration and prediction in nine water stage gauging stations are presented in Tables 5 and 6, respectively.

Obviously, the results of hydraulic simulations were fine, which validated the effectiveness of the hydraulic model. However, the effectiveness of the hydraulic model was not the emphasis of this paper but was introduced in the third series paper. Here, this

section was to make the implementing processes of this case study more complete. The simulated and observed outflows in 2007 and 2008 from the Three Gorges Reservoir are also compared in Figure 12, which shows good simulation performance. The relative error was less than 0.7, and the NSE for both was over 0.95. Here, the simulated outflow from the Three Gorges Reservoir was the discharge after simulation of the two types of HFUs mentioned above. Overall, the model gives good and reliable simulation results not only in the certain watersheds but also in the whole Three Gorges Reservoir area according to the outflow from the reservoir (Figure 12). Therefore, it can be concluded that modeling with a specific hydrologic or hydraulic method corresponding to certain HFUs is an effective and significant means for basin modeling.

**Table 5.** Observed and simulated water level and simulation efficiency of 22 flood events in 9 hydrological stations for model calibration.

| Water Stage Gauging Station | Flood Event | Flood Peak Level (m) | | | NSE |
|---|---|---|---|---|---|
| | | Observed | Simulated | Error | |
| Cuntan | 070731 | 176.8 | 177.6 | 0.8 | 0.87 |
| | 070827 | 172.7 | 172.6 | −0.1 | 0.95 |
| | 070902 | 174.6 | 174.7 | 0.1 | 0.88 |
| Changshou | 070731 | 166.4 | 165.7 | −0.7 | 0.94 |
| | 070903 | 162.3 | 161.4 | −0.9 | 0.87 |
| Qingxichang | 070731 | 159.4 | 159.4 | 0.0 | 0.97 |
| | 070827 | 152.3 | 151.9 | −0.4 | 0.90 |
| | 070903 | 154.4 | 154 | −0.4 | 0.93 |
| Zhongxian | 070731 | 152.9 | 152.5 | −0.4 | 0.91 |
| | 070828 | 148 | 147.9 | −0.1 | 0.99 |
| | 070903 | 149.2 | 149.3 | 0.1 | 0.96 |
| Wulong | 070730 | 194.5 | 194.2 | −0.3 | 0.98 |
| | 070902 | 178.7 | 178.5 | −0.2 | 0.94 |
| | 070909 | 179.9 | 179.2 | −0.7 | 0.95 |
| Wanxian | 070801 | 151.2 | 150.3 | −0.9 | 0.90 |
| | 071031 | 156.1 | 156.2 | 0.1 | 1.00 |
| Fengjie | 070801 | 149.8 | 148.8 | −1.0 | 0.84 |
| | 071031 | 156 | 156 | 0.0 | 1.00 |
| Wushan | 070801 | 148.9 | 148.2 | −0.7 | 0.82 |
| | 071031 | 156.1 | 156 | −0.1 | 1.00 |
| Badong | 070801 | 146.9 | 146.6 | −0.3 | 0.82 |
| | 071031 | 156 | 155.9 | −0.1 | 1.00 |

**Table 6.** Observed and simulated water level and simulation efficiency of 28 flood events in 9 hydrological stations for model validation.

| Water Stage Gauging Station | Flood Event | Flood Peak Level (m) | | | NSE |
|---|---|---|---|---|---|
| | | Observed | Simulated | Error | |
| Cuntan | 080724 | 173.5 | 173.4 | −0.1 | 0.90 |
| | 080812 | 175.5 | 176.1 | 0.6 | 0.90 |
| | 080929 | 174.9 | 174.7 | −0.2 | 0.90 |
| | 081105 | 175.9 | 176.1 | 0.2 | 0.95 |
| Changshou | 080424 | 157.1 | 156.9 | −0.2 | 0.94 |
| | 080724 | 161.6 | 160.9 | −0.7 | 0.93 |
| | 080812 | 175.5 | 176.1 | 0.6 | 0.90 |
| | 080929 | 162.8 | 161.8 | −1.0 | 0.90 |
| | 081105 | 174.6 | 174.1 | −0.5 | 1.00 |
| Qingxichang | 080425 | 155.0 | 155.1 | 0.1 | 0.98 |
| | 080708 | 152.3 | 152.1 | −0.2 | 0.97 |
| | 080724 | 154.5 | 154.4 | −0.1 | 0.96 |
| | 081106 | 173.9 | 173.7 | −0.2 | 1.00 |
| Zhongxian | 080425 | 154.3 | 154.4 | 0.1 | 1.00 |
| | 080817 | 150.9 | 150.7 | −0.2 | 0.98 |
| | 081106 | 173.4 | 173.3 | −0.1 | 1.00 |
| Wulong | 080724 | 181.3 | 181.4 | 0.1 | 0.95 |
| | 080817 | 182.2 | 182.1 | −0.1 | 0.87 |
| | 080904 | 183.1 | 182.9 | −0.2 | 0.93 |
| Wanxian | 080426 | 154.1 | 154.1 | 0.0 | 1.00 |
| | 080817 | 149.6 | 149.1 | −0.5 | 0.95 |
| | 081104 | 172.8 | 172.8 | 0.0 | 1.00 |
| Fengjie | 080425 | 154.0 | 153.9 | −0.1 | 1.00 |
| | 081104 | 172.7 | 172.6 | −0.1 | 1.00 |
| Wushan | 080425 | 154.0 | 153.8 | −0.2 | 0.99 |
| | 081104 | 172.6 | 172.5 | −0.1 | 1.00 |
| Badong | 080425 | 153.8 | 153.7 | −0.1 | 0.99 |
| | 081104 | 172.5 | 172.3 | −0.2 | 1.00 |

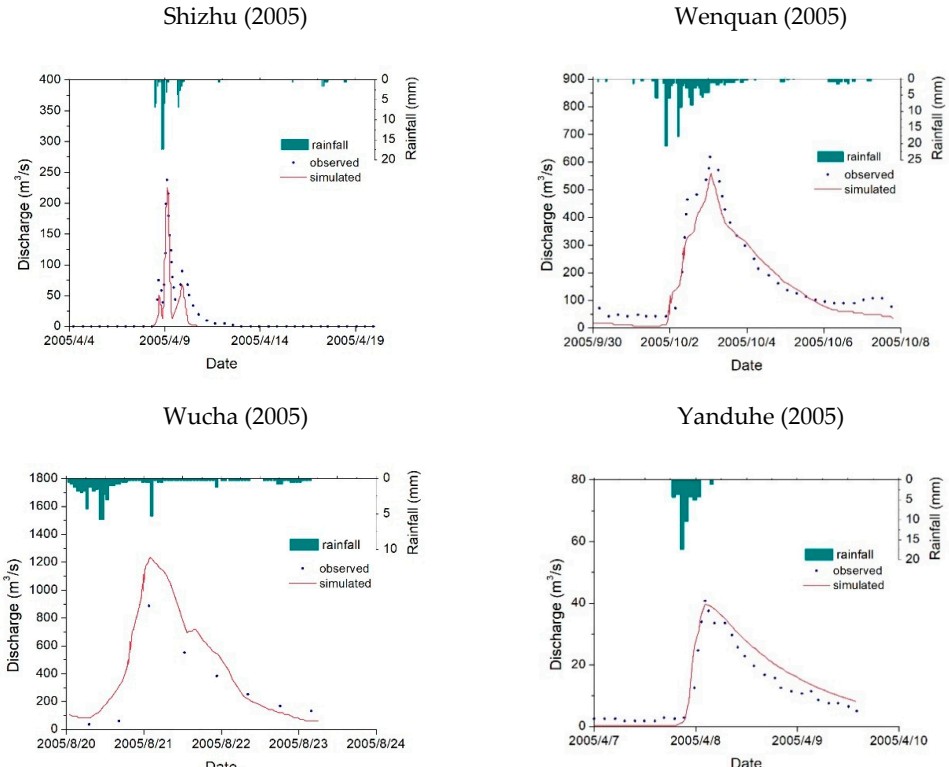

**Figure 10.** Rainfall distribution and simulated and observed discharge timeseries at four representative hydrological stations in 2005.

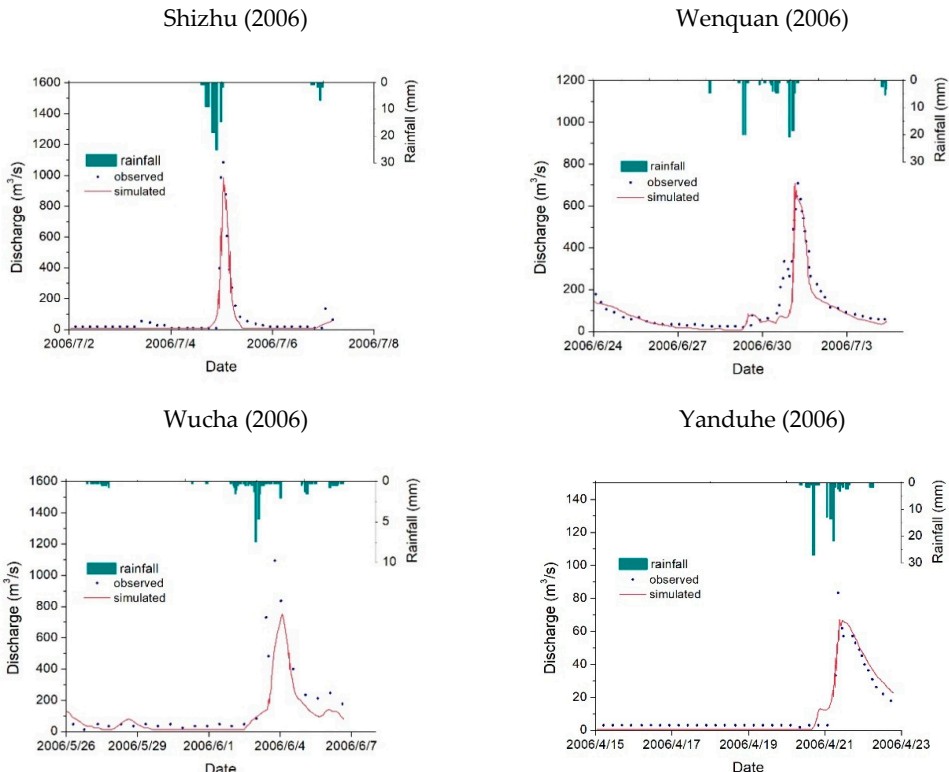

**Figure 11.** Rainfall distribution and simulated and observed discharge timeseries at four representative hydrological stations in 2006.

outflow from reservoir (2007)　　　　outflow from reservoir (2008)

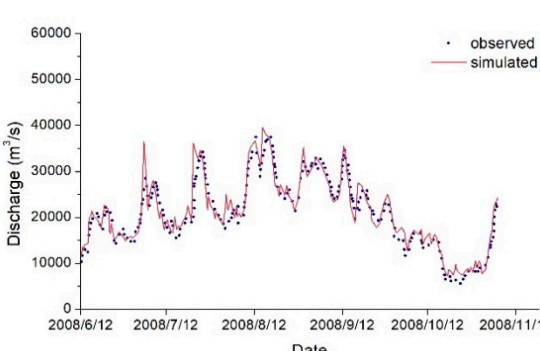

**Figure 12.** Comparison of simulated and observed outflows (m$^3$/s) from the Three Gorges Reservoir in 2007 and 2008.

## 4. Conclusions

The distributed-framework hydrologic modeling system (DF-HMS) was introduced by a structure diagram with different HFUs and hydrological processes. HFUs were classified into three categories, and hydrologic HFUs were emphasized in this paper. The digital basin generation model was developed based on the DEM to determine the essential information for hydrologic (hilly sub-watershed HFUs) and hydraulic (plain river HFUs) simulation at the basin scale. One case was applied in the Three Gorges area, which is a typical hilly area. The hydrologic simulation results are efficient when comparing observed discharge data in the Three Gorges area, where the NSE of discharge was larger than 0.8 for more than 91.6% of the studied flood events, and RE was smaller than 10% for more than 88.2% of cases. To some extent, the DF-HMS is a comprehensive system that includes various possible HFUs that will be applied in the next series papers. With the development of hydrologic models, HFUs also can be added and improved to fit into various hydrological circumstances.

**Author Contributions:** The work was conducted by G.C., W.H., X.F., C.W., X.L. this paper was written by G.C., and X.F., C.W. reviewed and improved the manuscript with comments; the data compilation and statistical analyses were completed by all authors. All authors have read and agreed to the published version of the manuscript.

**Funding:** This research has been financially supported by the National Key Research and Development Program of China (2018YFC1508200), Project 41901020 supported by NSFC, the Fundamental Research Funds for the Central Universities (B200202030), and Hydraulic Science and Technology Program of Jiangsu Province (2020003).

**Institutional Review Board Statement:** Not applicable.

**Informed Consent Statement:** Not applicable.

**Data Availability Statement:** Not applicable.

**Conflicts of Interest:** The authors declare no conflict of interest.

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
