# Peer review of "Distributed-Framework Basin Modeling System: II. Hydrologic Modeling System"

_water, doi:10.3390/w13050744_

Round 1

Reviewer 1 Report

The authors did a good job at responding to my comments. No concerns.

Author Response

We appreciate your work and help to improve our paper.

Reviewer 2 Report

I truly believe that the 4 papers could be synthesized into just one single paper, but I’ll leave that decision up to the Editor. Having said this, the authors have addressed each of my concerns and they are applauded for improving the papers. I appreciate the contribution of the authors in reporting their hydrologic modeling framework and I encourage them to work in future implementations with public repositories with codes and tutorials written in English for easier access to the international community.

Author Response

Thanks again for your comments. We appreciate your review and help with our paper. We have discussed with the editor and get the permission from the editor to publish these four papers after revision. We are also working on the code publication and tutorials written in English. The DFBMS has been developed for more than decades, it is a sophisticated modeling system (framework) so that it will take some time and effort to complet above tasks.

Reviewer 3 Report

The authors of the manuscript “Distributed-Framework Basin Modeling System: Hydrologic 2 Modeling System (Ⅱ)” made major alteration of their initial manuscript and have revised it according to the reviewers’ comments. The new output is an ameliorated version of the first manuscript. I recommend the major revision of the manuscript based on the following comments:

General comments:

  1. The new revised version of the manuscript should be submitted including the track changes to better understand the changes. It is very difficult to understand where the alterations and amendments within the manuscript are located.
  2. The responses on the authors should clearly mention the lines of the alterations, otherwise the paper could be misjudged. This means that for each response the exact lines within the revised manuscript should be provided.

Specific comments:

Line 48: “Hydrological processes in areas with different circumstances should…”. What do the authors mean by circumstances? I suggest the word “conditions”.

Line 65.: The same comments as previous. Replace the word “circumstances”.

Lines 94-97: The authors state in the manuscript” According to different mechanisms of runoff generation and/or movement, the distributed-framework basin modeling system (DFBMS) includes 11 kinds of hydrological feature units (HFUs) (first series paper).” In the exactly next sentence, the authors state “Each HFU has the same mechanism for runoff generation and/or movement.” As it is written, the 2 sentences are in controversy between them. The 2 sentences should be rephrased.

Lines 134-136: “Some HFUs may partly change to other HFUs in special conditions such as extreme weather, flood events, seasonal variation, artificial diversion”. What do the authors mean by the expression that some HFUs may partly change to other HFUs? In which way the HFUs change, since HFUs are defined as an area/region that has the same mechanism of runoff generation and/or movement (This definition is given in the 1st Paper). The authors should provide more details.

Lines 140-141: “…or we can attempt to build canopy interception and evapotranspiration models by constructing CO2 cycle patterns”. What do the authors mean by the expression “we can attempt to…”? When and where are you going to attempt to build….? There is no clear meaning, thus please rephrase.

Lines 143-144. “Snowmelt was considered where it existed and calculated with a related hydrologic method.”. According to the authors’ response to my similar comment in the initial review this is not true. In their reply, the authors clearly state: “The related hydrologic method or model for snowmelt has not been developed in the system because snowmelt was rarely happened in most basins we studied and we have never come across any research program involving snowmelt. But we do know its importance in special area and proposed corresponding HFUs. Also, we provide interfaces in the system for model program modules of snowmelt from other researchers to access.”

Thus, the authors should clearly mention (as they did few lines above with the SHE model for the evapotraspiration and the canopy interception) the following “The related hydrologic method or model for snowmelt has not been developed in the system since no relative necessity was met. However, the infrastructure for integrating snowmelt processes, such as the commonly used degree-day method (Skoulikaris et al., 2020), has been developed.”

Where: Skoulikaris, C.; Anagnostopoulou, C.; Lazoglou, G. Hydrological Modeling Response to Climate Model Spatial Analysis of a South Eastern Europe International Basin. Climate 20208, 1. https://doi.org/10.3390/cli8010001

Lines 144-145: “In the lake and reservoir HFUs, flood districts like paddy fields are also included when needed.” This is not clearly. Do the authors mean that paddy fields could be included because there is the proper infrastructure within the HFU, or do the users have the option to select the paddy fields option? Please be more precise.

Figure 3. The figure must be re-created. No legends, scale bars, place names, the location of the dams, compass are provided. The authors must also correlate the figure with the Figures 7 and 8, i.e. to use the same placenames.

Line 361. Section 2.2.3: The authors present as an example the Taihu basin. However, no one can understand where the Taihu basin is located. Is it part of the Three Gorges basin?  and if yes where inside the basin?  For which reason the authors selected the specific basin? More information are required by the authors. Is this basin connected with the section 3.1 Study Area and Corresponding Hydrological Characteristic Unit?

Line 583: precipitation stations instead of precipitation station. Please make the correction.

Line 589: case study instead of study case. Please make the correction.

Figure 7b and Figure 9. The relation of the two figures in not obvious. Different names-placenames are refereed to each figure. Please make the relation to be understood the relation of the two networks.

Lines 600-601. “For the plain river HFUs, the discharge of reservoir outflow in this study means a total discharge, and the outlet of channel reservoir is at dam site”. For which reason the authors do not just say total discharges and dam? It is very confusing to understand the meaning of the sentences using the proposed by the authors expressions when reading the rest of the manuscript.

For example in Figure 12, the reservoir outflow is what? And where?

Lines 601-602: “so the reservoir regulation doesn’t have to be considered during flood season here.” It is not clear why in flood seasons the reservoir regulation is not considered. Please be more precise for the reasons behind this argument.

Lines 604-605: “The measure discharge of reservoir outflow was from the stream gauging station downstream”. What is the meaning of this sentence, it cannot be understood? By changing the “discharge of reservoir outflow” to total discharge, as the authors suggest, then there is again no meaning.

Tables 5 and 6. What is the meaning of  the “Flood event” code in the Tables?

Author Response

Thank you for your comments and help to improve our series paper. Please check the attachment Word file for the replies.

Round 2

Reviewer 3 Report

The authors took into consideration all the reviewers comments and the revised version of the manuscript is at the proper level for being published. Congratulation to the authors.

This manuscript is a resubmission of an earlier submission. The following is a list of the peer review reports and author responses from that submission.

Round 1

Reviewer 1 Report

General comments

As a continuation from the 1st manuscript, the authors within this 2nd manuscript demonstrate information about a model that they developed, namely digital basin generation model, and about 2 (maybe 3?) HFUs, namely hilly sub-watershed and plain overland flow HFUs. The methodology for the digital basin generation model, for producing the basin characteristics, such as watershed area, runoff concentration path etch, with a use of DEMs, puts emphasis on the depressed pits, which is an actual problem with DEMs. This part of the manuscript is well presented and documented. The plain overland runoff generation HFUs are also well presented and documented in comparison with other HFUs.

What is of great concern is that the authors do not clearly state which model or models are available within the DFBMS, as well as the models that attributed for each HFUs. The authors should present a comprehensive list of the models per HFUs. The only case they do that, is in the Plain overland runoff generation HFUs. I also concluded, but it is not clearly stated, that for the Hilly sub-watershed HFUs the authors use the  Xinanjiang model. At any case, the authors apart from giving the aforementioned list of models, they should provide short descriptions about the models (input data, requirements on parameters etc). They proceed on simulations without giving specific information about the models.

The methodology and the outputs of the implementation process of their model to the case study area are presented without providing adequate information for evaluating both the methodology and the results. Authors should pay more attention on the way they present the implementation of the proposed new model. This paper has significant flaws which need to be radically ameliorated before being accepted.

Specific comments

Line 30. The authors should add very well-known references about the hydrological models, and say few words about the hydrological modelling itself, the types of models etc. Examples of literature:

Beven, K. J. (1990). A discussion of distributed hydrological modelling. In Distributed hydrological modelling (pp. 255-278). Springer, Dordrecht.

Wurbs, R. A. (1998). Dissemination of generalized water resources models in the United States. Water International, 23(3), 190-198.

Singh, V. P., & Woolhiser, D. A. (2002). Mathematical modeling of watershed hydrology. Journal of hydrologic engineering, 7(4), 270-292.

Lines 57-67. The aim of this 2nd manuscript should be presented in more clear way. I would propose the lines that refer to hydrologic model system (HMS) to be transferred in the previous paragraph. The digital basin generation model should be also added within the aim of the manuscript.

Line 78. The authors should also state some info about the DF-RMS, as they did with the DF-HMS

Lines 89-91. “…(HCUs) such as sub-watershed, sloped planes for overland flow, river cross-sections, and grids, which were explained in the first series paper.”. This is not correct. UCUs are just mentioned with limited justification and information. Thus, I would propose more details about HCUs since they are important structural objects of the model.

Lines 101-104. “Some existing methods like related calculations in the SHE model … evapotranspiration models by constructing CO2 cycle patterns [17].” Which one is used within the authors’ model? Provide more information of the so called “simplified coefficient method”.

Lines 105-106. “Snowmelt was considered where it existed and calculated with a related hydrologic method.” Which is the related hydrologic method? Snowmelt simulation is not just a method. Provide more information about your statement, and inform which methods you use within your model.

Lines 111-119. The authors present literature information about extracting major drainage path networks, catchment area etc. Do they authors use the methods that are presented? If yes it should be clearly stated.

Line 122. SFD and MFD. Please provide information about these abbreviations.

Line 122. A reference is needed for the specific statement.

Lines 126-127. “Therefore, the SFD algorithm was selected for the DFBMS described here.” Which is this SFD algorithm. The authors should clear say which one they use.  

Line 131. “2) Identify the steepest downslope for every cell and accumulating catchment area downslope along the runoff concentration paths connecting adjacent cells.” It’s not really clear what you mean. Please explain this step with a more clear way. 

Line 136. “Calculate the geographic and hydrologic parameters for the sub-watershed.” Which are the hydrologic parameters? The slope? Please be more precise or rephrase.

Line 232. The authors state that 3 typical HFUs were described in detail: hilly sub-watershed, hilly river, and plain overland flow HFUs. However, in the abstract they say that only 2 HFUs are described. Please make the appropriate corrections.

Line 250. Is the Xinanjiang model integrated within the DF-HMS?

Line 255. Why the authors use the threshold of one hour? What do you exactly mean by “temporal scale of discretization”?

Lines 263-264: “the net rainfall is calculated according to precipitation input and calculation of canopy interception.” Are these 2 calculation conducted within the DF-HMS? If yes please provide some more details.

Lines 424-426.” The river flow model includes one-dimensional and two-dimensional models to simulate for single channels and the river network as well as the coupling process between different models.” What do you mean by the “river flow model”? Do you mean the model that is integrated within the plain river HFU? Which are the one-dimensional and two-dimensional models that you use?

Lines 446-447: “For this kind of HFU, the traditional conceptual hydrological model is mostly applied.” Which are these methods, and which methods are available within the model?

Line 476: “For the flood forecasting model…”. What is the flood forecasting model?

Line 474. Model Calibration and Validation. It’s not clear which model the authors use for the hydrologic simulation. Do the authors use exclusively the Xinanjiang rainfall-runoff model for the simulation of hilly sub-watersheds? The authors just present the outputs with remarkable increased correlation for such a large basin.

Table 2. What is the DC column?

Lines 506-508: “Here the discharge of reservoir outflow in the Three Gorges Reservoir was the discharge after simulation of two HFUs mentioned above.” Apart from not accurately presenting the way that hilly sub-watershed and plain river HFUs were simulated, I cannot also really understand the way that the authors achieved such a good correlation between their models outputs and the outflows of the dam! Dams have specific operational rules that depend on water availability, seasonality, water demands etc, and these rules have a dominant role on the dams’ outlets. In which way the authors were able to simulate the operation of the dam? We have a fragmented river which does not follows the rules of simulating natural watersheds. The statement of the authors cannot be received as accurate without presenting specific information on the simulation process.

Lines 506-508: “Here the discharge of reservoir outflow in the Three Gorges Reservoir was the discharge after simulation of two HFUs mentioned above.” The way that plain river HFUs were simulated is not provided.

Reviewer 2 Report

The subject of the manuscript „Distributed-Framework Basin Modeling System: Hydrologic Modeling System (Ⅱ)” is current and very important. The authors of the series of paper present a new modelling approach which simulates the hydrological process at basin scale. In the present work it was presented a new modelling approach the hydrological feature unit (HFU). The article requires a lot of improvement:
- Eleven different HFUs are presented, however the models that are interconnected in each HFU are however not adequately represented and commented upon. It should be noted that there is no clear overview of the work on this issue (maybe infographic, table or diagram).
- The methodology and model verification are not clearly described.
Also, model validation issues should be more clearly described in the manuscript.
- Was it not better to choose a smaller catchment area as a case study? What was the criterion for choosing this and not another case study?
- The work lacks an in-depth analysis and discussion of the results, which is a serious shortcoming in the scientific article.
- References from positions 18 to 28 are incorrectly described.
- The literature review, although not extensive, does not include the most recent, up-to-date works. Out of 38 references, only 6 are from the last 10 years. The rest are much older. This needs to be improved.
Summing up, I think that the work, although interesting, should be properly corrected and supplemented.

Reviewer 3 Report

This series of 4 papers presents the description of a distributed basin modeling system composed of several components. The strength of the model laid on the flexibility and number of processes that can be integrated and modeled across the hydrologic and hydraulic components. In general, the papers are well-written, and the methods included in the hydrologic and hydraulic components are well presented. Having said this, I do not consider proper to present this work as a series of 4 papers, the overall structure looks more suitable for a dissertation document or a chapter book, but the presented format does not fit with the overall goal of a scientific paper, in which should maximize the synthetizes of methods, results, and discussions without losing accuracy and transparency, which reminds me the popular said: “I didn't have time to write a paper, so I wrote a book.” I encourage the authors to reconsider to condense the work into one single paper in order to show their wonderful work. Below, I’m describing my major and minor comments for all 4 papers.

[Major Comments]

  1. The 4 papers should be presented as a single paper. “Paper 1” can be easily synthesized as the introduction section, “Paper 2” and “Paper 3” is the section method, and “Paper 4” would be the Case Study. I noted through the papers several redundancies that can be avoided in order to achieve the best synthetizes in the work. For instance, “Paper 2” and “Paper 3” show a Case study; however, that should be the main purpose of “Paper 4”. There are several sections in “Paper 2” and “Paper 3” that can be moved to a Supplemental Material section (See Minor Comments).
  2. As I mentioned before, the strength of the study is found in the hydrologic and hydraulic components. However, “Paper 4” decreases the overall impact of the presented model. The authors were limited to show that the proposed model was able to replicate the discharge, water depth in some gauges just for a short period of time (calibration 1 year, validation 1 year). In general, there is not an analysis of the spatial distribution of the model performance, and there is no understanding of how the different model components, either hydrologic or hydraulic, is improving the representation of the hydrologic processes. This paper shows a new hydrologic model framework, therefore, should be expected to find an extended analysis of the different capabilities of the models showing the improvement of the model with and without different components
  3. The authors did not provide the source code or repository of the model. This is essential for future implementations of the model in the hydrologic community. In the case that the model is not available to the public, the authors must provide further details about the configuration in computational times used to run the simulations.

Minor Comments on: “Distributed-Framework Basin Modeling System: Overview and Model Coupling (I)”

[Line 25] What advantages? This statement is ambiguous

[Line 33] What does FH69 stand for?

[Line 106] Be careful using the argument of “Temporal GIS”, this is a matter of perspective, somebody could argue that including time-series to represent rainfall fields is sufficient to be in the realm of “Temporal GIS”. Besides, geological models do not seem an appropriate example to show the inconvenience of temporal representations in the current hydrological models, note that the geological processes evolve in a dramatic larger time scale in comparison to hydrologic processes explored in most of the hydrologic models.

[Figure 5] The captions need to be improved

Minor Comments on: “Distributed-Framework Basin Modeling System: Hydrologic Modeling System (II)”

[Line 20] Only two HFUs? what about the other 9 HFUs? Is there any model documentation?

[Line 122] What do SFD and MFD stand for?

[Section 2.1 2.1.1, 2.1.2] This section could be omitted or summarized in one or two paragraphs. Most of this content may be considered as general knowledge in the hydrologic community (e.g. estimation of flow direction D8), so there is no need to be so explicit in its development. Another option is included as Supplemental Material.

[Equation 2] What water depth and Chezy coefficient are used in Eq 2? Are they varying over space? Or is it just the DEM elevation used as the water depth in this case? Or are assumed constant across all the DEM processing?

[Lines 261-268] The runoff generation process and the overland flow must be explained in this section! The authors are just limited to provide some references; this is one of the key elements in the description of any hydrologic model.

[Lines 270-278] Be specific with the hydrograph method. This statement is vague, what equations and approximations are the authors using for the hydrograph routing method?

[Section 2.2.3] So does the plain overland runoff generation considers land use, but the Hilly- subwatersheds do not? Section 2.2.3 is nicely documented, however, section 2.2.1 is poorly described.

[Line 320] What specific parameter range? Be specific.

[Line 323] How necessary is to include this complexity in modeling runoff on the paddy fields? Have the authors provided any evidence of the adequacy in including this process? This should be explored in high detail in “Paper 4”

[Line 361] Provide ranges for Hp, Hu, and Hd

[Section 2.3] What about the modeling in woodland land use?

[Section 2.2.4] Include a description of the overland runoff method used. Again, what about the woodland surface?

[Section 2.3] This is not necessary if it is mentioned in “Paper 3”

[3 Study Case] This should be part of “Paper 4”

[Section 3.2] The evolution of the model performance needs to be improved. Please consider using the Nash–Sutcliffe model efficiency coefficient since has been used as a standard in the hydrologic community

[Section 3.2] Be specific in how the calibration was performed. What method? And what parameters were calibrated in this case study?

[Section 3.2] What was the computational time?

Minor Comments on: “Distributed-Framework Basin Modeling System: Hydraulic Modeling System (III)”

[Section 2] Large part of this paper could be included as an Appendix or Supplementals Information

[General] The equation numbering is incorrect, please verify.

[Lines 42-59] If there is no further discussion about these aspects through the paper, then this section should be removed.

[Section 2.4] This should be part of the “Paper 4”

[Line 443-Line 444] Rewrite “1982 cases...” for “case for the year 1982”; same for 1991.

[Figure 10] It is a better option to use a color bar to show the velocity field

Minor Comments on: “Distributed-Framework Basin Modeling System: Application in Taihu Basin (IV)”

[Figure 11] Are there only 4 streamflow gauges? Show statistics RMSE, Nash–Sutcliffe model efficiency

[Section 3.2] Why is the validation period and calibration period so short? I assume that there should be longer streamflow records within the basin, however, the authors only used one year for calibration and one for validation which obscures the true overall model performance that could be achieved with larger hydrologic records.

[General] Show the drainage area associated with each streamflow station

Reviewer 4 Report

This second series paper mainly describes DF-HMS which is a major component of DFBMS. The authors did a case study in the Three Gorges area and showcased good model performance in terms of streamflow calculation. Overall, good experiment design and excellent model results. I have some minor concerns as followed:

1. Line 475, is 4-year data sufficient for your calibration and validation? should you need longer term data? This may depend on data availability but it seems to me only 4-year data might not be sufficient.

2. Can you explain more about why the model doesn't perform well for large (>1000 cms) and small (<100 cms) flows? For flood prediction, it's very crucial to get peak flow right (especially for large cms).

3. In addition to RE and R^2, can you also calculate RMSE?